# Interactions between Diffuse Light and Cucumber (*Cucumis sativus* L.) Canopy Structure, Simulations of Light Interception in Virtual Canopies

**Yingyu Zhang** [1], **Juan Yang** [1], **Marinus van Haaften** [2,3], **Linyi Li** [1], **Shenglian Lu** [4], **Weiliang Wen** [5], **Xiuguo Zheng** [1], **Jian Pan** [6] and **Tingting Qian** [1,*]

1   Institute of Agricultural Science and Technology Information, Shanghai Academy of Agricultural Sciences, Shanghai 201403, China; yingyu.zhang@saas.sh.cn (Y.Z.); yangjuan@saas.sh.cn (J.Y.); lly@saas.sh.cn (L.L.); zhengxiuguo@saas.sh.cn (X.Z.)
2   Inholland University of Applied Sciences, Domain Agri, Food and Life Sciences, Rotterdamseweg 141, 2628 AL Delft, The Netherlands; marinus.vanhaaften@inholland.nl
3   TU-Delft, Technology, Policy and Management, Jaffalaan 5, 2628 BX Delft, The Netherlands
4   College of Computer Science and Information Technology, Guangxi Normal University, Guilin 541004, China; lsl@gxnu.edu.cn
5   Beijing Key Lab of Digital Plant, National Engineering Research Center for Information Technology in Agriculture, Beijing 100097, China; wenwl@nercita.org.cn
6   School of Agriculture and Biology, Shanghai Jiao Tong University, 800 Dongchuan Road, Minhang District, Shanghai 200240, China; nilllice@sjtu.edu.cn
*   Correspondence: qiantingting@saas.sh.cn; Tel.: +86-150-0075-3513

**Abstract:** Plant photosynthesis and biomass production are associated with the amount of intercepted light, especially the light distribution inside the canopy. Three virtual canopies (n = 80, 3.25 plants/m$^2$) were constructed based on average leaf size of the digitized plant structures: 'small leaf' (98.1 cm$^2$), 'medium leaf' (163.0 cm$^2$) and 'big leaf' (241.6 cm$^2$). The ratios of diffuse light were set in three gradients (27.8%, 48.7%, 89.6%). The simulations of light interception were conducted under different ratios of diffuse light, before and after the normalization of incident radiation. With 226.1% more diffuse light, the result of light interception could increase by 34.4%. However, the 56.8% of reduced radiation caused by the increased proportion of diffuse light inhibited the advantage of diffuse light in terms of a 26.8% reduction in light interception. The big-leaf canopy had more mutual shading effects, but its larger leaf area intercepted 56.2% more light than the small-leaf canopy under the same light conditions. The small-leaf canopy showed higher efficiency in light penetration and higher light interception per unit of leaf area. The study implied the 3D structural model, an effective tool for quantitative analysis of the interaction between light and plant canopy structure.

**Keywords:** 3D structural model; light interception; plant simulation; virtual cucumber canopies; diffuse light; canopy structure

## 1. Introduction

Cucumber (*Cumcumis sativus* L.), responding like a semitropical plant, prefers growing conditions of high temperature, humidity, and light intensity [1]. Recent studies indicate that cucumbers are sensitive to sufficient oxygen and inorganic nutrient solutions [2,3]. The growing and production of cucumbers in greenhouses provides multiple advantages in terms of controlling growth conditions and crop maintenance techniques, which ensures crop productivity [4,5]. The production cost in greenhouses is much higher than in fields [6]. This gives the growers a more urgent need to maximize the crop yield to cover the higher production cost of the greenhouse. Greenhouse growers attempt to increase their yields by managing the interaction between the plant and its environmental factors in the greenhouse. Knowledge of the relation between the phenotype-genotype of plants and their interaction

with its environment (e.g., $CO_2$, light, temperature), leads to important breeding and growing applications [7–10]. This study focuses on investigating the interception and distribution of light in cucumber canopies

Light has short-term and long-term influence on plants. In the short term, light intensity and the amount of light interception by the canopy are associated with plant photosynthesis and biomass production [11,12]. In general, a 1% of reduction in solar radiation reduces the average production of cucumbers by 0.8–1% [13]. In the long term, light affects the leaf area, distribution of dry matter, and leaf orientation increasingly [14–16]. Light has more influence in a larger crop due to a larger leaf area index (LAI) [17,18]. However, leaf formation can also be enhanced under a higher light intensity, which leads to a larger leaf area [19,20]. This long-term interaction between light and morphology indicates more biomass production through such morphological changes due to an increased light interception, which consequently increases light-use efficiency [10].

In addition to the amount of incident light falling on the canopy, the light distribution also affects crop production. The harvestable proportion of cucumbers can increase by 4.3% in kilos under a diffuse covering material [21]. When the incident light is applied at the top of a closed canopy, the irradiance in the canopy decreases with increasing depth [22–24]. Diffuse light penetrates deeper into a plant canopy compared with direct light, which further increases photosynthesis and crop production [21,25,26]. The seasonal patterns of crop production have also been simulated by converting all direct light to diffuse light, indicating a production increase in cucumber of up to 4.0% in summer [27].

Multiple methods and models are applied to estimate the light interception under different assumptions, where the leaf area is used. For instance, projected leaf area can be used to estimate the intercepted light if the incident light is direct light with an orientation perpendicular to the ground [28]. Lambert-Beer's law assumes that leaves of the canopy are uniformly distributed in space to attenuate light in a vertical direction [29], while cucumbers in China were normally cultivated in double rows, which results in heterogeneous distribution of leaves. Thornley et al. [30] estimated the light propagation for a whole canopy by splitting the total amount of light into fractions along three Cartesian directions. These assumptions, based on theoretical models, to some extent, simplify the situation; they restrict the scope and depth of studies in light interception in terms of micro-climate (e.g., light gradient) of the canopies and the environmental difference on the organ level [31].

Limitations in the estimation of light interception exist due to the lack of information given by phenotypic data. The manual that collected phenotypic data usually has a small sample size with lower accuracy and precision [32]. The plants' three-dimensional (3D) structural model explicitly considers the static state of individual leaf surfaces [28,33–35]. A model can compare the difference in light distribution and light interception of plants [6,28,35], similar to the penetration of direct sunlight of cucumber plants, which has been investigated with a structural model [36]. Likewise, the effect of plant density and plant distribution on light interception of canopies with different plant spacing distances was investigated [28,37]. Studies that include the condition of diffuse light are rare [31,38]. Additionally, this study provides a more quantitative perspective of light interception using 3D structural models. The aim of this study is to analyze the light interception and distribution in cucumber canopies with a static 3D structural model based upon data derived from cucumber plants.

## 2. Materials and Methods

### 2.1. Plant Material

The cucumbers (*Cucumis sativus* L.) were cultivated in the greenhouse at Shanghai Jiaotong University (31°7′12″ N, 121°22′48″ E). The planting date was 1 September 2020. There were 224 cucumber plants in a Recombinant Inbred Lines (RILs) population with different genotypes for each plant cultivated. The population had different plant phenotypes regarding leaf size and plant size due to the segregation of the character leaf size. The plants were grown in the soil with a high-wire system in a double row. The greenhouse was a Venlo-type and covered by glass with a transmissivity of 82%. Fans were applied on

the roof between the rows of plants to provide ventilation (Figure A2). There were 10 rows with 1-m intervals in the compartment. The distance between two adjacent plants was 40 cm (Figure A3). The plant density was 3.25 plants/m$^2$. Plants were vertically trained and twisted around the cord. The plants were cultivated under a sufficient supply of water and nutrients. The plants were not wilting nor dehydrated on the day of experiment. During the experiment, which took place from 1 September to 22 September 2020, no side shoots were observed before or on 22 September. The average day temperature and night temperature was 29.02 °C and 23.27 °C, respectively. The average relative humidity was 44.82% during the day and 38.87% during the night. The $CO_2$ level was maintained via ventilation through a window opening, where the atmospheric $CO_2$ level was approximately 400 ppm. On 22 September, when the digitization of plants was conducted, the plant canopies were not closed and still in their vegetative stage.

### 2.2. Simulation of 3D Structural Model

2.2.1. Data Collection and 3D Structural Digitization

The digitization of all cucumber plant samples (N = 224, no border plant included) was based upon 224 real plants described in Section 2.1 on 22nd September 2020, before the first harvest of fruit. A 3D tracking system (Fastrack NS-1016, Polhemus, Colchester, VT, USA) was used to collect 3D architectural data of each cucumber plant in the greenhouse compartment. During the measurements, the plants were not moved along the high wire, in order to achieve a higher accuracy in digitizing the natural state of the cucumber plant and to measure the azimuth angle and leaf angle more accurately.

The plant parts were digitized according to a sequence defined in Figure 1. The morphological parameters data was obtained from after digitization, included internode length (IL, cm), petiole length (PL, cm) leaf length (LL, cm), and leaf width (LW, cm). For each node, one point was placed at the petiole insertion on the stem (p1), followed by an insertion point of the lamina (p2). There were four points digitized from each lamina (p2–p5). The visualization of 3D plant structures was produced by 'Visualization and 3D Reconstruction of Plant Models', a software developed by Shanghai Academy of Agricultural Sciences (Figure 2) [31]. The program filled the prepared leaf-shaped samples into the skeleton [31,38].

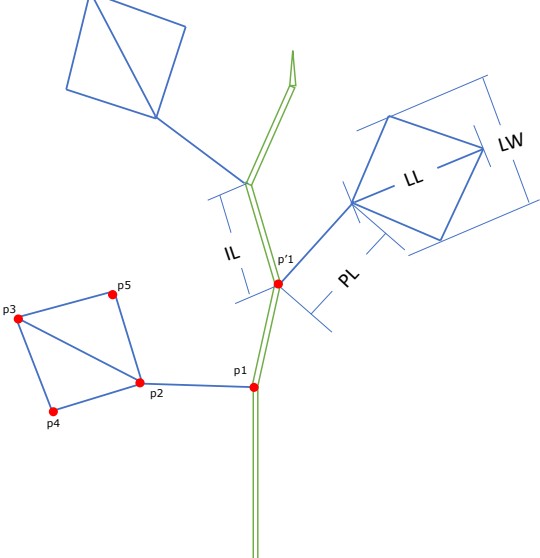

**Figure 1.** Example of the measurements of architectural parameters. p1–p5 show the positions of digitization of a phytomer unit of cucumber and its leaf geometry in a defined sequence. p'1 indicates the start point of the next measured phytomer. Each phytomer consists of four architectural parameters: LW represents the leaf width (cm); LL represents the leaf length (cm); PL represents the petiole length (cm); IL represents the internode length (cm).

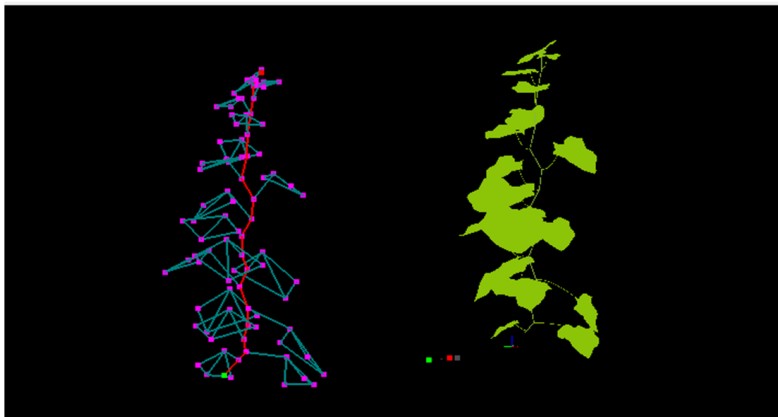

**Figure 2.** The skeleton model (**left**) and a virtual model filled with leaf shaped samples (**right**) of a cucumber plant from digitized data produced by the software program "Visualization and 3D Reconstruction of Plant Models".

2.2.2. The Simulation Settings

The structure of real plants can reveal more realistic simulation of light interception grown in greenhouses with sufficient structural diversity [31]. The 3D-model was simulated with three virtual canopies with different leaf sizes (small, medium, big) under three different scenarios of light conditions, which resulted in nine simulation scenarios. The simulation sessions between canopies with different leaf sizes and under different ratios of direct/diffuse light are conducted to compare the efficiency of light interception.

- Three virtual canopies

After digitizing and dividing the 224 plants into three groups with different leaf lengths (see Figure 3), three virtual canopies were modelled, one canopy for each group individually. From each group of digitized plants, 20 were selected randomly and included in the 3D-plant model for that group. The individual 360° 3D plant model of each of the 20 plants in each group was divided into 5 different angles to enable a multiplication of distinctive virtual plants with distinctive canopies. The number of virtual 3D-modelled plants in each of the 3 groups was 80 and 240 in total.

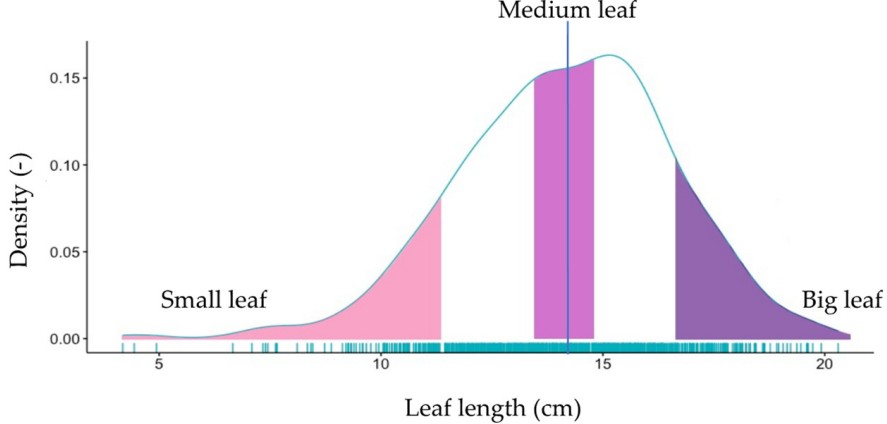

**Figure 3.** The distribution of leaf length of the plant samples (averaged for each plant) measured on 22 September. Three plant groups (small, medium, big; n = 20) of digitized plant 3D models were randomly selected for reconstruction of virtual canopies based on leaf length. The vertical line indicates the average leaf length (=14.1 cm).

The leaf area of a single leaf was estimated by a multiplying a factor of 0.743 with the product of the leaf length and the leaf width [39].

The cultivation characteristics of the virtual crop (n = 240) was similar to the cultivation characteristics of the real plants during the cultivation: zonally a 1-m interval, plant distance of 40 cm and a plant density of 3.25 plants/m$^2$. In order to eliminate the influence of border effects, only the center plants (n = 6) were included in the analysis of light interception of each canopy.

As shown in Figure 4, the small-leaf canopy had the smallest number of leaves, the smallest individual leaf area, and the shortest plant height. The leaf area index (LAI, cm$^2$ leaf area/cm$^2$ soil) of the 3 virtual canopies were 0.19, 0.47 and 0.89 respectively.

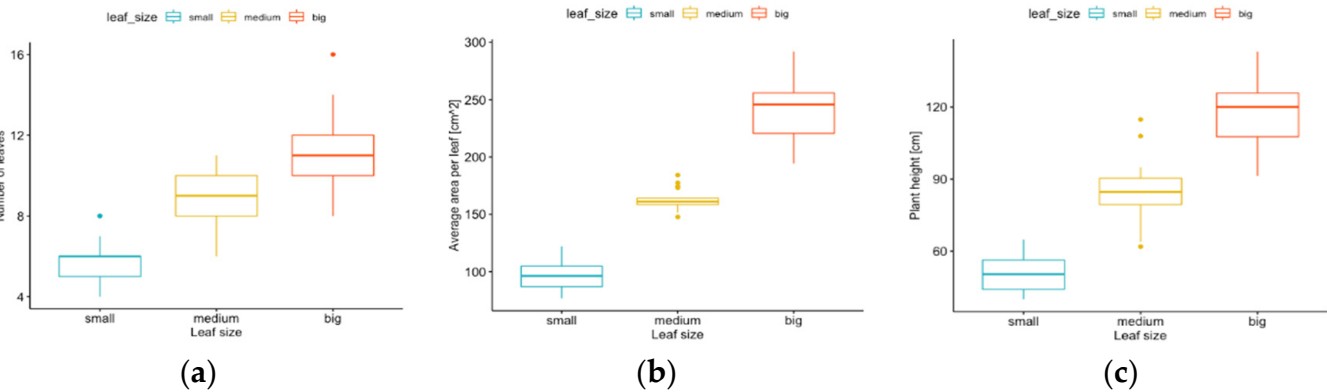

**Figure 4.** The general description of three plant groups with different leaf sizes (small, medium, big) (**a**) number of leaves; (**b**) individual leaf area averaged per plant (cm$^2$); (**c**) plant height (cm) for three groups. The leaf area (cm$^2$) is estimated by leaf length (cm) × leaf width (cm) × 0.743.

- The scenarios of light conditions

Three scenarios of light conditions were set based on the radiation dataset (Table 1). The total radiation and the ratio of diffuse light were obtained from the daily meteorological dataset of basic meteorological elements of the China Meteorological Radiation International Exchange Station (1957–2020) (The website for the Daily meteorological dataset can be obtained from: http://meteor.ckcest.cn/mekb/?r=data/detail&dataCode=RADI_MUL_CHN_DAY (accessed on 27 April 2021)), in Shanghai. Scenario 1 (s1) has the lowest ratio of diffuse light with the highest total radiation, representing a sunny condition. Scenario 2 (s2) has a medium ratio of diffuse light and total radiation, representing a medium light condition. Scenario 3 (s3) has the highest ratio of diffuse light with the lowest total radiation, representing a cloudy light condition. Though the ratio of diffuse light was increased by 221.6% in s3 compared to s1, the amount of total radiation was reduced by 56.8% in s3 compared to s1. The indoor incident PAR in Table 1 was converted from total radiation manually. The conversion was based on the assumption that the incident photosynthetic active radiation (PAR), of which the wavelength was 400–700 nm, accounted for 47% of the global radiation of 20 MJ/m$^2$d [40]. The height of sunshine took the reference at 12 PM on the summer solstice in the lunar calendar (21 June 2019). The simulation of light interception was conducted by using the radiosity-graphic combined model (RGM, with PC configurations: Windows 10, CPU of AMD3950x, RAM of 64G). RGM was used to simulate direct light and diffuse light separately for the 3D structural model [41]. The leaf was assumed to absorb 87%, transmit 7%, and reflect 6% of PAR [25]. The simulated results were illustrated in the light interception for each leaf (with labelled numbers) in units of μmol/s.

Due to the different totals of radiation in three scenarios of light conditions, the effect of diffuse light could be partly constrained. Therefore, the total radiation in the three scenarios (s1, s2, s3) was normalized to be the same value by transforming the total radiation of the scenarios into 65.61 MJ/m$^2$ (the sum of radiation in s1, s2 and s3), while keeping the ratios of diffuse light unchanged, which resulted in three scenarios of normalized radiation (sn1, sn2, sn3). After the normalization, the total radiation in each scenario became dimensionless

and the differences in total radiation were canceled out. Three scenarios of normalized radiation (sn1, sn2, sn3) were used to compare the single effect in light interception caused by diffuse light.

**Table 1.** The scenario setting for the simulation of light interception.

| Scenarios | Date | Total Radiation (MJ/m$^2$) | Indoor Incident PAR (μmol/m$^2$s) | Ratio of Diffuse Light |
|---|---|---|---|---|
| s1 | 3 June 2019 | 29.48 | 426.41 | 27.8% |
| s2 | 9 June 2019 | 23.38 | 336.16 | 48.7% |
| s3 | 10 June 2019 | 12.75 | 183.18 | 89.6% |

*2.3. Data Analysis*

The graphical visualization of statistical data was arranged in Microsoft Office Excel 2019 and RStudio version 1.2.5033. The visualization of virtual canopies was displayed by MeshLab v2020.12. Normality of the data was tested using a Kolmogorov–Smirnov test (Table A1). Possible differences in the light interception between the three different simulated scenarios and the three different groups of leaf size were tested with an ANOVA. Differences between the three different groups of leaf size and between the three scenarios were tested with a pairwise comparison Tukey Contrast in the 'glht' function in the 'multicomp' package of RStudio version 1.2.5033.

The relationship between cumulative light interception and cumulative leaf area (from top to bottom) was estimated with a linear regression. A simple linear regression (the slope *k* and the intercept *b*) was calculated by the simple linear regression with the function 'lm()' in RStudio version 1.2.5033.

After the execution of linear regression in R for six scenarios of light condition (s1, s2, s3, sn1, sn2, sn3) and three virtual canopies (small, medium, and big leaf), the slopes (*k* for s1, s2, and s3 and kn for sn1, sn2, and sn3) and intercepts (*b* for s1, s2, and s3 and bn for sn1, sn2, and sn3) of a total of 18 combinations (3 × 6 = 18) were further analyzed in boxplots. Theoretical definitions were given to the slopes and intercepts:

- the slopes (*k* and *kn*) represented the efficiency of light penetration into the canopy.
- the intercepts (*b* and *bn*) indicated either the projected leaf area or total incident radiation depending on the comparison.

Estimating the results of linear coefficient estimates in Excel 2019 provided exact similar statistics.

**3. Results**

In this section, the interaction between ratios of diffuse light and canopy structure were simulated. In Section 3.1, the simulated results of light interception were analyzed to explore the effect of leaf sizes and light conditions. After normalizing the radiation, the effect of diffuse light was either compared solely or in combination with the difference in incident radiation. The analysis was conducted using the center plants (n = 6) of each canopy to simulate the performance of light interception of plants inside a canopy in reality. Section 3.2 showed the analysis of the relationship between light interception per plant from the top layer to the bottom and the corresponding leaf area. Firstly, the light interception on the canopy level was compared among three plant groups with different leaf sizes. Then the light interception of three canopies was compared among three scenarios with different ratios of diffuse light.

*3.1. The Simulated Results of Light Interception*

3.1.1. Leaf Size

The light interception of plant groups with different leaf sizes was significantly different for at least one group compared to the other groups (*p* = 0.00578, F value = 5.416). The average of the simulated light interception was estimated on a plant level. According

to the results shown in Figure 5a, the big-leaf canopy had 56.2% more light interception compared to the small-leaf canopy. Moreover, the variation in light interception of each plant was larger in the canopy with big leaves. The smaller variation in light interception in the small-leaf canopy resulted in a better light penetration in the canopy. The small-leaf canopy had less cumulative light interception due to the fewer leaves and shorter stems, while the big-leaf canopy had more light interception (and larger variation) due to a higher leaf-density and taller stems.

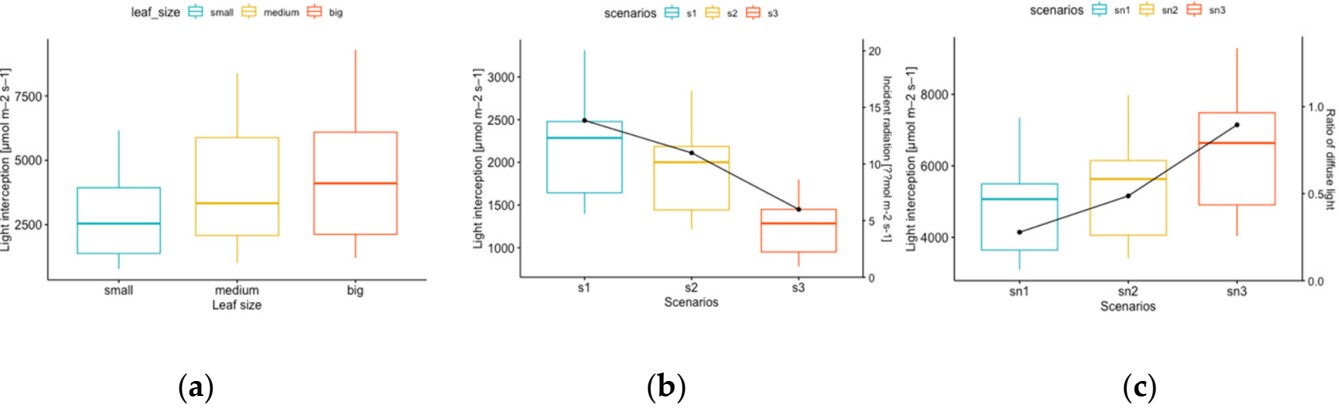

(a)       (b)       (c)

**Figure 5.** (**a**) Boxplot of light interception ($\mu$mol s$^{-1}$) per plant among groups with different leaf sizes (small, medium, big); (**b**) boxplot of light interception ($\mu$mol s$^{-1}$) per plant under different light conditions, with the line indicating the incident radiation ($\mu$mol m$^{-2}$ s$^{-1}$); (**c**) boxplot of normalized light interception ($\mu$mol s$^{-1}$) per plant under different light conditions, with the line indicating the ratio of diffuse light (compared to total incident radiation), ranged from 0–1. Scenarios include s1 (27.8% of diffuse light, 426.41 $\mu$mol/m$^2$s of incident PAR), s2 (48.7%of diffuse light, 336.16 $\mu$mol/m$^2$s of incident PAR), and s3 (89.6%of diffuse light, 183.18 $\mu$mol/m$^2$s of incident PAR); the scenarios of normalized incident radiation (sn1, sn2, and sn3) have the same ratios of diffuse light as s1, s2, and s3, respectively.

### 3.1.2. Scenarios with Different Light Conditions

In Figure 5b, the plants intercepted 26.8% more light in scenario 1 (s1) than scenario 3 (s3), which had the least incident radiation and higher ratio of diffuse light. The incident radiation of s3 was much weaker than s1 and scenario 2 (s2), as indicated by the line according to the y axis on the right side in Figure 5b. This corresponded with the Tukey pairwise comparison of light interception of the three scenarios. From the Tukey post hoc test it was shown that the light interception in s3 was significantly smaller than in s1 or in s2 (both $p < 0.001$), meanwhile there was no significant difference between the light interception between s1 and s2 ($p = 0.162$). In the three scenarios of normalized radiation (sn1, sn2, sn3), the incident radiation was assumed to be the same. The simulated result of scenarios of normalized radiation was presented in Figure 5c, where the trend of light interception among scenarios was the opposite of that in Figure 5b. The interception of sn3 was 34.4% more than sn1. The ratio of diffuse light of sn1 was lower than sn3, as indicated by the line according to the y-axis on the right in Figure 5c. This indicated that the light interception was influenced by both the amount of incident radiation and ratio of diffuse light. The amount of incident radiation, however, outperformed the amount of diffuse light ratio due to an increased light interception of 26.8% in s1 more than s3, with 56.8% increased radiation and 226.1% reduced diffuse light.

### 3.2. The Light Interception vs. Leaf Area

Regardless of the leaf size, the number of leaves and total leaf area also affected light interception. In this section, the relationship between light interception and leaf area was investigated. As an example, the linear relationships between light interception of the big leaf group and the corresponding leaf area were illustrated in Figure 6a with $R^2 > 0.9$. Similarly,

the incident radiation was normalized and lead to the linear relationships presented in Figure 6b. This relationship expressed the difference in light interception not only among the three scenarios (Figure 6a,b), but also among the canopies with different leaf sizes (Figure 7).

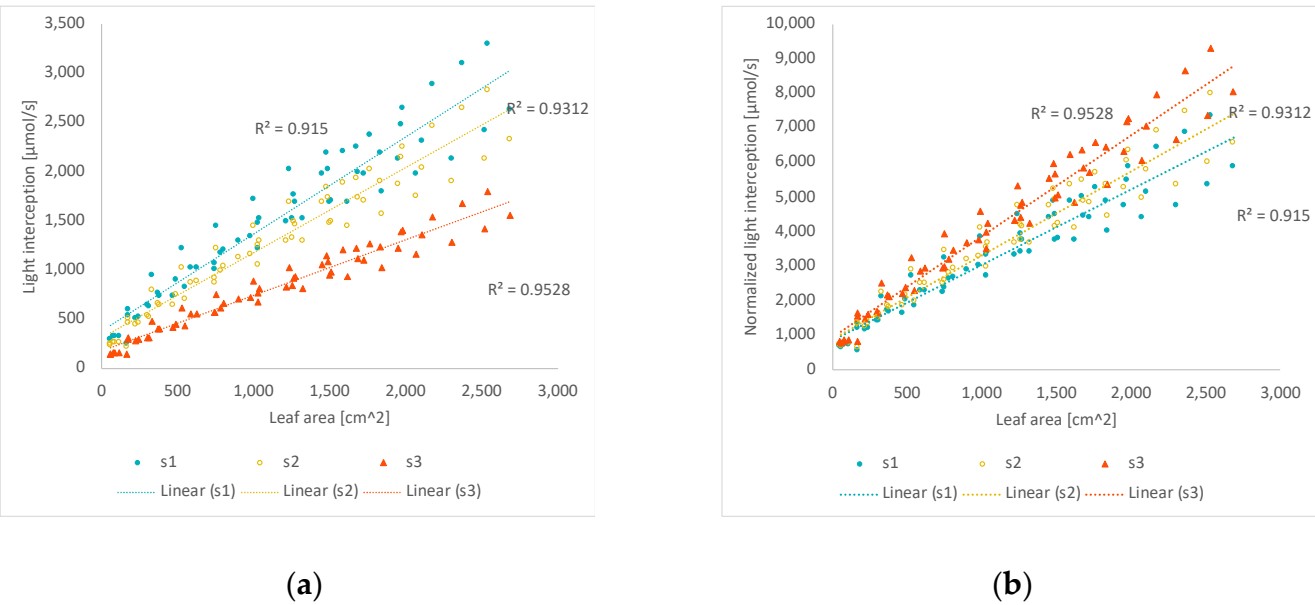

(**a**)                                                                                   (**b**)

**Figure 6.** (**a**) Scatterplot of light interception (µmol s$^{-1}$) against the leaf area (cm$^{-2}$) for the big leaf group under different light conditions (s1, s2, and s3). (**b**) Scatterplot of normalized light interception (µmol s$^{-1}$) against the leaf area (cm$^2$) for the big leaf group under different light conditions (s1, s2, and s3). The light interception and leaf area were cumulated from the top of the plant to the lowest, by layers. Each scenario of light condition was fitted linearly, resulting in an R-square (R$^2$). Scenarios included s1 (27.8% of diffuse light, 426.41 µmol/m$^2$s of incident PAR), s2 (48.7%of diffuse light, 336.16 µmol/m$^2$s of incident PAR), and s3 (89.6%of diffuse light, 183.18 µmol/m$^2$s of incident PAR).

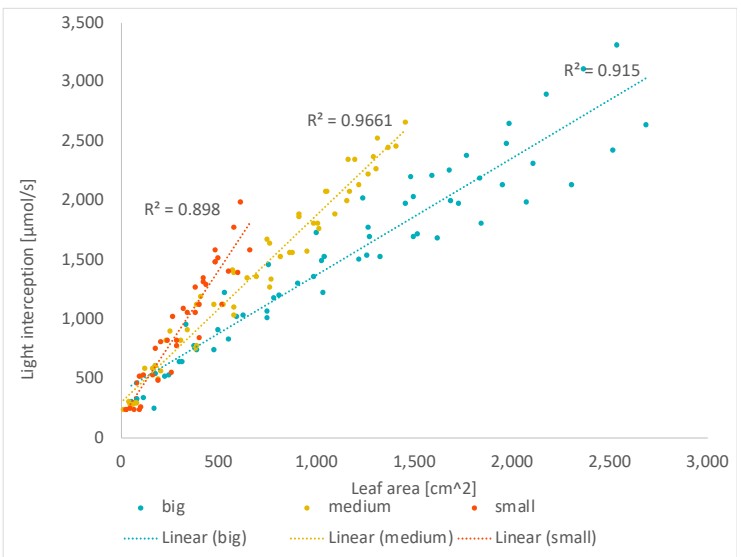

**Figure 7.** Scatterplot of light interception (µmol s$^{-1}$) against leaf area (cm$^2$) of three groups with different leaf sizes (big, medium, small) within scenario 1 s1 (27.8% of diffuse light, 426.41 µmol/m$^2$s of incident PAR). The light interception and leaf area were cumulated from the top of plant to the lowest by layers. Each scenario of light condition was fitted linearly with R-squared (R$^2$).

The slope ($k$) and intercept ($b$) of the linear regression ($y = kx + b$) of the relationship between light interception and leaf area are shown in Table 2. The parameters regarding the normalized light interception ($kn$, $bn$) and $R^2$ were also included. The $R^2$'s remained unchanged after the normalization of incident radiation. The slopes ($k$ and $kn$) indicated the efficiency of light penetration into the canopy. The intercepts ($b$ and $bn$) represented the incident radiation just above the top layer of the canopy, which further indicated the projected leaf area.

**Table 2.** The parameters of the fitted equation ($y = kx + b$, $x$ = leaf area, $y$ = light interception). The slopes ($k$ for s1, s2, and s3 and $kn$ for sn1, sn2, and sn3) and intercepts ($b$ for s1, s2, and s3 and $bn$ for sn1, sn2, and sn3) of a total of 18 combinations were displayed with 4 decimals. The $R^2$'s were indicated in the last column.

| Leaf Size | Scenarios | $k$ | $kn$ | $b$ | $bn$ | $R^2$ |
|---|---|---|---|---|---|---|
| big | 1 | 0.9878 | 2.1909 | 382.97 | 849.40 | 0.9150 |
| big | 2 | 0.8633 | 2.4287 | 315.70 | 888.19 | 0.9312 |
| big | 3 | 0.5667 | 2.9257 | 180.79 | 933.43 | 0.9528 |
| medium | 1 | 1.5774 | 3.4985 | 301.17 | 667.98 | 0.9661 |
| medium | 2 | 1.4000 | 3.9386 | 251.47 | 707.48 | 0.9692 |
| medium | 3 | 0.9402 | 4.8542 | 146.52 | 756.50 | 0.9736 |
| small | 1 | 2.4971 | 5.5385 | 167.17 | 370.76 | 0.8980 |
| small | 2 | 2.1881 | 6.1558 | 154.43 | 434.46 | 0.8909 |
| small | 3 | 1.4338 | 7.4029 | 109.30 | 564.33 | 0.8706 |

### 3.2.1. Total Leaf Area

As depicted in Figure 8a, the small leaf group had a larger slope than the medium and big leaf groups. This indicated a higher efficiency of light interception into the canopy, as well as a higher light interception per unit leaf area. The three slopes ($k$) of the three plant groups differed significantly ($p = 0.0225 < 0.05$) from each other. After the normalization of incident radiation, the difference in slopes ($kn$) was more significant ($p = 0.00169 < 0.01$, Figure 8c). In Figure 8b, the small leaf group had a smaller intercept than the medium and big leaf groups, indicating a smaller projected leaf area within the same soil surface. The difference in intercepts ($b$) among plant groups was not significant ($p = 0.134 > 0.1$). According to Table 2, the intercepts ($b$ or $bn$) of the same scenario indicated the degree of canopy closure, or the projected leaf area of a single plant. The intercept ($b$) of the same plant group indicated the difference in incident radiation, so s3 had the smallest intercept ($b$) in each plant group. After the normalization of incident radiation, the intercept ($bn$) of the small leaf group was significantly ($p = 0.00064 < 0.01$) smaller than the medium and big leaf group (Figure 8d).

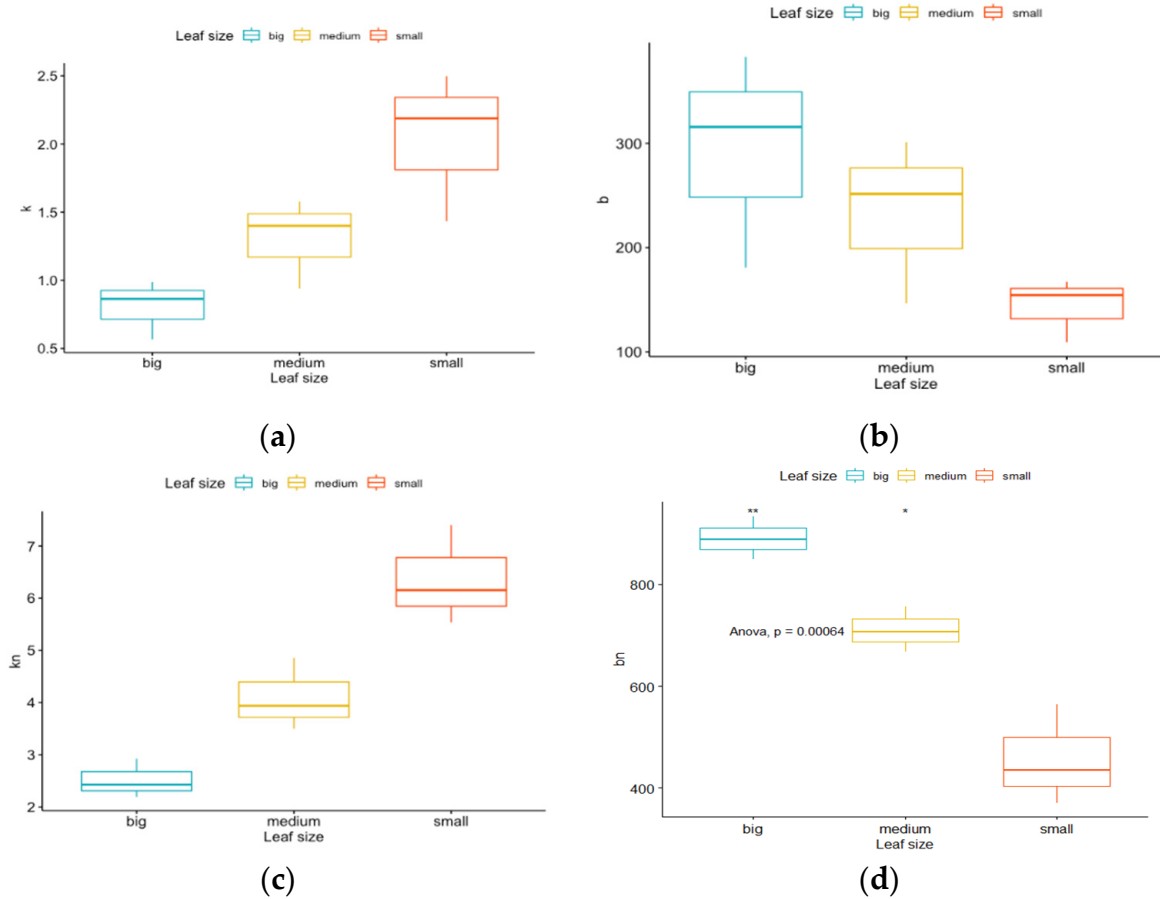

**Figure 8.** The parameters of the relationship ($y = kx + b$, $x$ = leaf area, $y$ = light interception) between the light interception and leaf area cumulated from the top of the plant to the lowest by layers, fitted by the simulated result of the center plants (n = 6). (**a**) the boxplot of the slopes $k$ among big-leaf, medium-leaf, and small-leaf canopies with different incident radiation and ratios of diffuse light (s1, s2, s3). (**b**) the boxplot of intercepts $b$ among big-leaf, medium-leaf, and small-leaf canopies with different incident radiation and ratios of diffuse light (s1, s2, s3). (**c**) the boxplot of slopes $kn$ among big-leaf, medium-leaf, and small-leaf canopies with the same incident radiation and ratios of diffuse light (sn1, sn2, sn3). (**d**)the boxplot of intercepts $bn$ among big-leaf, medium-leaf, and small-leaf canopies with the same incident radiation and ratios of diffuse light (sn1, sn2, sn3). Scenarios include s1 (27.8% of diffuse light, 426.41 μmol/m$^2$s of incident PAR), s2 (48.7% of diffuse light, 336.16 μmol/m$^2$s of incident PAR), and s3 (89.6% of diffuse light, 183.18 μmol/m$^2$s of incident PAR); the scenarios of normalized incident radiation (sn1, sn2, and sn3) have the same ratios of diffuse light as s1, s2, and s3 respectively.

### 3.2.2. Ratios of Diffuse Light

According to Table 2, the slopes ($k$) of the scenarios of the plant groups were insignificant ($p = 0.427 > 0.1$), indicating that s1 didn't have a higher efficiency in light interception inside the canopy than s2 and s3. After the normalization of incident radiation, the slopes ($kn$) of s3 became larger than s1 and s2, though the difference was still insignificant ($p = 0.714 > 0.1$), indicating an absence of any trend in $k$.

Since the intercepts ($b$) of the same plant group indicated the difference in incident radiation ($p = 0.183$), the intercepts ($bn$) of the same plant group among scenarios became smaller (Table 2). This smaller difference was observed from Figure 8b,d which displayed that the variation of intercepts among the plant groups became smaller after the normalization. This explained that after the normalization of incoming radiation in the three scenarios, the difference in intercepts ($bn$) within the same plant group was even less significant ($p = 0.797 > 0.1$).

## 4. Discussion

This study contributed to the literature by presenting the results of light interception simulated in a 3D structural model. Studies on the interception and distribution of light interception, especially diffuse light, were limited. This section discussed the results on light interception in relation to other studies, possible limitations of using this model, and future possibilities to simulate light interception of crop canopy.

### 4.1. The Effect of Radiation

The simulated light interception with the same incident radiation showed that canopies could intercept diffuse light better than direct light. This corresponded with the study of Hemming on the effect of diffuse light, which indicated that an increase of the ratio of diffuse light could increase the photosynthesis of the canopy, and consequently increased the production [21]. Because diffuse light was scattered in all directions, it could penetrate deeper into a plant canopy compared to direct light. With a higher ratio of diffuse light, the variation of light interception per plant was smaller (Figure 5c. The incident radiation (direct light) was set to be perpendicular to the ground, which meant that the top layer of the projected leaf area was the most active in intercepting direct light. The leaves inside the canopy were shaded and unable to intercept direct light, which resulted in a larger variation of light interception per plant, or even per leaf under direct light (Figure A1).

Although the ratio of diffuse light was increased by 221.6% in s3 rather than s1, the results showed that the total light interception of s3 was less than s1 (Figure 5b), because the amount of incident radiation was reduced by 56.8% in s3 compared to s1. Similar results were achieved for total light interception without normalization. Under natural light conditions, the increased proportion of diffuse light was usually companied by a reduction in the global radiation, which negatively influenced productivity [42]. Other studies also evaluated this effect on cloudy days with only diffuse light, and there was at most 30% of radiation at the top of atmosphere; meanwhile, a rather clear sky enabled more than 80% of radiation to reach the Earth's surface [40].

By means of diffuse light, a higher uniformity of light distribution could be realized [21]. This uniform light distribution helped realize equality in production. Otherwise, the shadows over the leaves caused by mutual shading and greenhouse construction could have a negative influence on plant production. Modern covering materials could help create more diffuse light. These materials contained pigments and macro- or microstructures that transformed all the incoming direct light into diffuse light, without significantly reducing light transmission [21].

Besides the scattering effect of radiation (diffuse light), the geometrical effect of radiation also influenced light interception [43]. Changes in the spatial geometry of radiation could result in a different outcome of light interception since leaves intercept light from another angle. In this study, the azimuth and elevation of the light source was perpendicular to the ground. However, the position of the sun changed during the day, which affected the area of leaf surface intercepting the light at different time points. Qian et al. simulated both circumstances (the sun at 12 pm and the moving pattern of the sun during a day) [38]. They found that the conclusion for the treatments were the same, regardless of if the sun position was fixed at 12 pm or changing over the day.

Kahlen simulated the 3D structural cucumber canopies with plant densities of 1 and 2 plants/$m^2$ followed by a greenhouse experiment in Germany [28]. The plant density (3.25 plants/$m^2$) in this study was relatively high for the simulation of light interception [44,45]. Optimizing plant density resulted in increased intercepted light, which further enhanced productivity [22,46]. Increasing plant density could modify the distribution pattern of the light within the canopy [46,47]. Higher plant density could also reduce both the proportion of incident light intercepted per plant and the red/far-red ratio of light at the bottom of the canopy, which also influenced tillering [48]. Moreover, the foliage density and arrangement could also influence the local variation of light interception [6].

### 4.2. The Impact of Leaf (Geometry and Area)

With a larger total leaf area (or LAI), the leaf surface to intercept light was larger. For instance, the small-leaf canopy had better light penetration with a larger k (Figure 8). However, the LAI of a big-leaf canopy was much larger, which led to higher total light interception [13]. Before the canopy closure, an increase in the number of leaves could contribute to an increase in light interception. Due to the effect of mutual shading, this pattern was eliminated after the canopy closure. It canceled out the increased leaf area under the direct light from above because the projected leaf area remained unchanged. The light penetration could be influenced by the interspace within the canopy, (the compactness and layers of leaves). Each layer of leaves could absorb 80–85% of PAR (photosynthetic active radiation, 400–700 nm) and reflect and transmit the rest, which is called scattering [49]. If the soil surface was covered by white plastic sheets, the reflection from the base would increase to 50–80% compared to 10–20% by bare soil, with a 7% increase in photosynthesis for LAI at 3 [43]. In addition, the lower side of a leaf could also reflect the light to the downward layers, depending on its thickness.

Besides the leaf area, the plant size also influenced the biomass partitioning to the fruit, which may refer to the cost of light interception. A variety of biological processes (e.g., metabolism and mass flow) could be meditated by the size of living organisms and organs [50]. At an organ level, an increased leaf size led to a larger cost in building and maintaining the same amount of leaf area for the plant [51]. Increased tree sizes resulted in an increased cost of light interception in terms of increased maintenance respiration, which reduced the available energy for stem growth [52]. Some studies mentioned that in cucumber plants, more biomass was partitioned to leaves and stems in order to intercept more light to assimilate supply, if the total biomass production was low [16]. This could result in less biomass partitioned to the fruit and more biomass partitioned to other parts, especially side shoots. Though the side shoots also produce leaves for more light interception, the biomass partitioned to the side shoots for extra stem growth and leaf growth increased the sink strength of vegetative parts.

### 4.3. The Impact of Extinction

The light interception was modelled based on the principle that increased leaf area enhanced light interception, while mutual shading inhibited it [43]. This principle resulted in the exponential extinction of light, which was incorporated into the law of Lamber–Beer [29]:

$$F_{int} = exp(-k_e{\cdot}L) \tag{1}$$

where $k_e$ ($k_e$ referred to light extinction coefficient, for distinguishing $k$ in Section 3.2) stood for the light extinction coefficient. $F_{int}$ stood for the fraction of light intercepted at the canopy depth of $L$ (expressed in overlying LAI). The range of typical values for $k_e$ was 0.5–0.8 [43].

Though $k_e$ was usually expressed as a constant value in the models, it was influenced by both the geometry of radiation and the geometry of leaf position and orientation (the azimuth angle and leaf angle). As discussed above, the changes in the geometry of radiation and that of leaf position and orientation caused differences in light interception. This effect could be expressed by a different value of $k_e$ in the function, for instance to express the different extinction profiles of the direct and diffuse light. When these extinction profiles were not measured specifically, the $k_e$ of diffuse flux was about 0.7 on average [43]. Moreover, a known $k_e$ could estimate the LAI from the Lambert–Beer function, with the radiation measured above and below the canopy [40]. With $k_e = 0.8$, LAI of 1 led to 55% light interception; LAI of 2 led to 80%, and LAI of 3 led to 90% [43]. In the assumption of Lambert–Beer's law to estimate light interception, the leaves were uniformly distributed in space to attenuate light in the vertical direction [29].

Unlike the static model used in this study, real plant leaves possibly adapted to the environment to intercept more light and avoid shading, due to changes in red/far-red ratio of light inside the canopy [33,53]. This kind of adaptation to the environment was

expressed as phenotypic plasticity [54]. Some of the short-term adaptations, like the leaf movement, needed to be recorded or simulated on the plant structure over a period of time. A dynamic model could estimate this kind of adaptation. Morphological changes of plants took time (weeks or months) before expressing the responses to the changes in the environment [55]. The dynamic models could facilitate the studies in phenotypic plasticity by studying these short-term plant responses. For the long-term adaptations, such as specific leaf area (SLA, the ratio between leaf area and leaf dry mass), which reflected leaf thickness and leaf size [54], multiple measurements for the same population (or the same cultivar) should be taken under different climate conditions [55]. By comparing the structure of plants measured under different climate conditions, the adaptation of plants to the environment could probably be further investigated.

In this study, a static 3D plant structural model was used for exploring the light interception on cucumber canopies. The 3D structural model offered opportunities to investigate the interaction between the environment, plant structure, and function [38]. Based on the 3D structure of plants, Ray Tracing technology and Radiosity technology, the simulation of light distribution accounted for: (1) the 3D spatial structure changes of the plant canopy, (2) the spatial distribution of plant organs, (3) surface optical characteristics, and (4) incident radiation [41]. In this way, the photosynthetic simulation based on light interception could be refined to a single organ or even a surface element scale [56]. There were also other existing models calculating radiation transfer used for predicting radiation absorption [57], canopy conductance [58], and evapotranspiration [59].

## 5. Conclusions

The 3D structural model is an effective research tool for quantitative analysis of the interaction between light and plant canopy structure. The findings of the study could offer new perspectives for model improvement in light interception, studies on plant plasticity, and breeding selection regarding the use of light based on the 3D plant structures. The influence on light interception referred to two aspects: radiation and canopy structure.

The results showed that a higher ratio of diffuse light is beneficial to light interception since it could better penetrate into the canopies. With a 226.1% increase in the ratio of diffuse light with the same incident radiation (PAR), the light interception increased by 34.4%. However, the 56.8% of reduced radiation caused by an increased proportion of diffuse light inhibited the advantage of diffuse light. The proof was indicated by a 26.8% of light interception reduction in the scenario with 226.1% more diffuse light in realistic light conditions.

Regarding the canopy structure, the linear relationship between cumulative light interception and cumulative leaf area (from top to bottom) implied a higher light penetration efficiency and higher light interception per unit leaf area in a small-leaf canopy via a larger slope of the linear regression. The big-leaf canopy had more mutual shading effects, but its larger LAI enabled the canopy to intercept 56.2% more light than the small-leaf canopy under the same ratio of diffuse light. A larger intercept in the linear regression of the big-leaf canopy could refer to a larger projected leaf area, which has more advantages under direct light.

**Author Contributions:** Conceptualization: Y.Z. and T.Q.; data curation: X.Z.; formal analysis: Y.Z.; funding acquisition: S.L. and T.Q.; investigation: J.P. and T.Q.; methodology: Y.Z.; project administration: J.Y. and L.L.; resources: J.P.; software: S.L. and W.W.; supervision: J.Y., L.L. and T.Q.; validation: T.Q.; visualization: Y.Z.; writing—original draft: Y.Z.; writing—review and editing: M.v.H. and T.Q. All authors have read and agreed to the published version of the manuscript.

**Funding:** This research was funded by the National Natural Science Foundation of China, grant number 61762013, the Science and Technology Foundation of Guangxi Province, grant number 2018AD19339 and the Shanghai Science and Technology Committee Program, grant number 21N21900700.

**Institutional Review Board Statement:** Not applicable.

**Informed Consent Statement:** Not applicable.

**Data Availability Statement:** The data presented in this study are available on request from the corresponding author. The data are not publicly available due to their containing information that could compromise the privacy of research participants.

**Acknowledgments:** The authors would like to thank Danfeng Huang for her insightful guidance, and Yeying Xu, Zhiwen Zhou, Minglu Tian, and Songtao Ban for their meaningful discussion.

**Conflicts of Interest:** The authors declare no conflict of interest.

## Appendix A

**Table A1.** *p*-values in the Kolmogorov–Smirnov Test for the traits with null-hypothesis of a normal distribution.

|  | Internode Length | Leaf Length | Leaf Width | Petiole Length |
|---|---|---|---|---|
| *p* value | 0.0827 | 0.2893 | 0.2822 | 0.9416 |

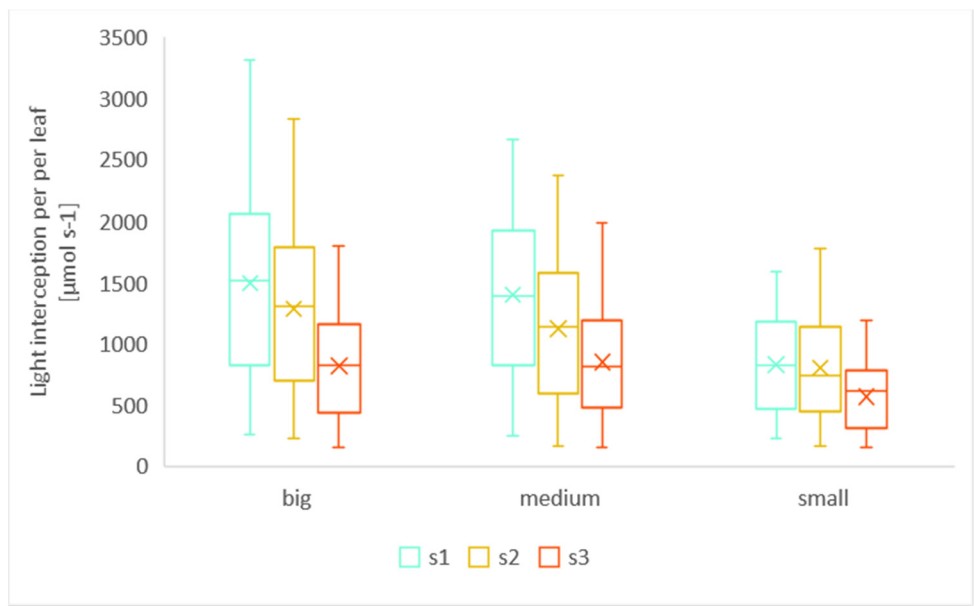

**Figure A1.** The simulated light interception per leaf for three plant groups (small leaf, medium leaf, and big leaf) and three scenarios (s1, s2, and s3).

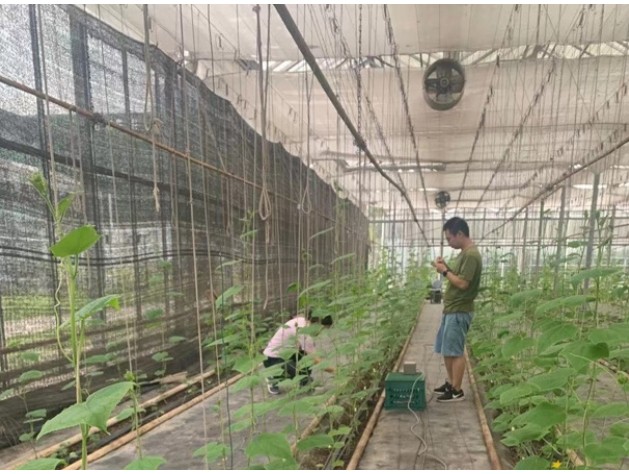

**Figure A2.** A picture of the experimental greenhouse, taken during the digitization of cucumber plants on 22 September 2020.

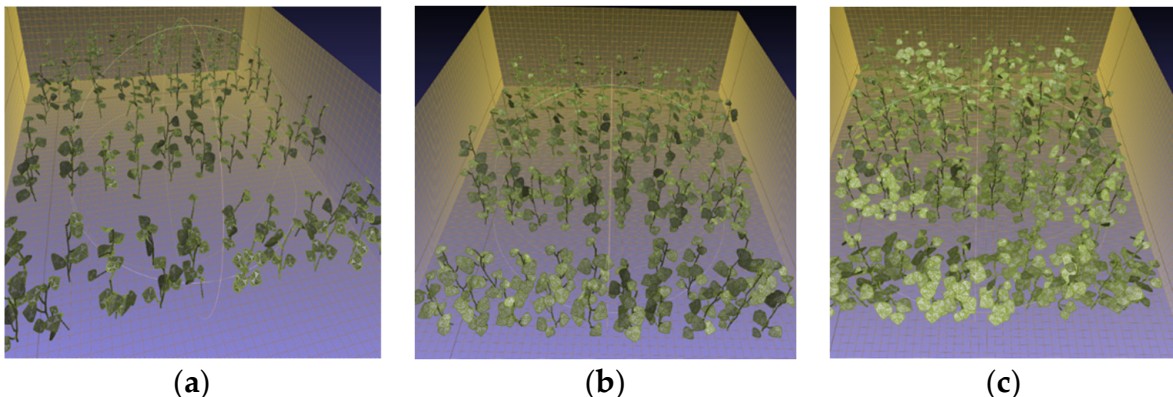

(a)          (b)          (c)

**Figure A3.** The preview of three virtual canopies (n = 80) with different leaf sizes. (**a**) small leaf; (**b**) medium leaf; (**c**) big leaf.

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
