# Peer review of "Interactions between Diffuse Light and Cucumber (Cucumis sativus L.) Canopy Structure, Simulations of Light Interception in Virtual Canopies"

_agronomy, doi:10.3390/agronomy12030602_

Round 1

Reviewer 1 Report

Some graphs look vague because of low resolution or small letters in axis label.

lines 138-144:
I have suggested to add the information (OS, CPU, RAM, etc.) of pc or workstation you used for the simulation.

lines 187-189:
I have suggested to add short explanation of RGM. How RGM treats the optical properties of plant leaves?

lines 468;
The authors have mentioned 'thickness of leaves'. The term of specific leaf area (SLA) may be appropriate for discussing the relationship between light interception and thickness of leaves. The discussion of SLA and the simulation result may enhance the value of this manuscript.

Author Response

Thank you very much for your insightful feedback.

Reviewer 2 Report

Zhang et al. presented a canopy light interception model. They reconstructed field cucumber canopy structure and explored the impact of different light conditions on radiation absorbance. The results are presented in clear logic and support the conclusions. My suggestion would be to improve the overall storyline. 

The authors seem to be ambiguous about the key message of the paper. I feel there are two key findings. First, the model is a useful tool for horticulture and particularly greenhouse set up. However, the authors did not mention anything about the application of the model. Moreover, it would be important to compare the model to existing models/methods of estimating plant light absorbance. Second, the model showed an potentially unidentified phenomenon-- the fraction of absorbed light (fAPAR) is lower when light intensity (PAR) is lower. As far as I know, most models assume the fAPAR to be a function of leaf area index and thus remain constant with varying PAR. It would be valuable for the authors to highlight these key findings and present the paper clearly defined goals. 

There are existing models that calculate radiation transfer. It would be great if the authors can acknowledge their existence and define the distinguishing feature of this model. 

Wang and Jarvis 1990 https://doi.org/10.1016/0168-1923(90)90112-J

Wang and Leuning 1998 10.1016/S0168-1923(98)00061-6

Luo et al. 2018 https://10.1002/2017JG003978

Details commons

L22-23: The logic of the sentence is unclear to me. What does ‘light interception increase more diffused light mean’?

L26: LAI is not defined yet. Also, not sure why mentioning LAI here while the rest of the paper focused on leaf area. 

L129: Figure 1 and Figure 2 left side are almost identical. Consider using only one of them.

L138: It is unclear to me what the model assumes for leaf absorbance and reflectance. Are those parameters measured or taken from literature. For this model, those are two key parameters?

L278-280: The logic between the two sentences is unclear to me. 

L473: The conclusion is too long and not well organised. Consider using five to six sentences to explain the importance of the study, the key findings and how the findings could be used. 

Author Response

(The authors gave the same response as above.)

Reviewer 3 Report

The aim of this study was to analyze the light interception and distribution in cucumber canopies with a static 3D structural model based upon data derived from cucumber plants.  

I made some comments in the attached file

Author Response

(The authors gave the same response as above.)
